# Incremental Causal Effect for Time to Treatment Initialization

**Andrew Ying**
Irvine, CA 92606, USA
aying9339@gmail.com

**Zhichen Zhao**
Department of Mathematics
University of California, San Diego
La Jolla, CA 92093, USA
zhz147@ucsd.edu

**Ronghui Xu**
Herbert Wertheim School of Public Health & Human Longevity Science,
Department of Mathematics and Halıcıoğlu Data Science Institute
University of California, San Diego
La Jolla, CA 92093, USA
rxu@ucsd.edu

## Abstract

We consider time to treatment initialization. This can commonly occur in preventive medicine, such as disease screening and vaccination; it can also occur with non-fatal health conditions such as HIV infection without the onset of AIDS. While traditional causal inference focused on 'when to treat' and its effects, including their possible dependence on subject characteristics, we consider the incremental causal effect when the intensity of time to treatment initialization is intervened upon. We provide identification of the incremental causal effect without the commonly required positivity assumption, as well as an estimation framework using inverse probability weighting. We illustrate our approach via simulation, and apply it to a rheumatoid arthritis study to evaluate the incremental effect of time to start methotrexate on joint pain.

## 1 Introduction

As a motivating example we borrow one from Bonvini et al. (2023) on behavioral health services for probationers, in order to reduce their chances of re-arrest. The question of interest, to be elaborated below, is about shifting the odds of receiving such a treatment. This can be achieved via affecting probationers' likelihood of attending services by, for example, providing transportation stipends. In public health, guidelines can be provided in order to affect people's likelihood of getting disease screening or vaccination.

An intervention such as in the above examples on the treatment distribution, is referred to as incremental intervention. The resulting causal effect of an incremental intervention is called the incremental causal effect (Kennedy, 2019; Naimi et al., 2021; Kim et al., 2021; Sarvet et al., 2023; Bonvini et al., 2023). In general the incremental causal effect has the interpretation of a policy effect on the population.

To better understand incremental intervention we provide a brief overview of the types of interventions in causal inference, which centers around answering "what happens if a particular treatment assignment changes ..." type of questions. An intervention on the treatment can change the value of a treatment, like from control to treated, or it can change its distribution, which is the case for incremental intervention. Causal inference literature started with investigating *static deterministic* interventions (Rubin, 1974; Hernán & Robins, 2020). Here 'static' is in contrast with 'dynamic', where the (possibly time-varying) treatment assignment may depend on covariates (and prior treatments and outcomes) (Robins, 1986; Murphy, 2003; Robins, 2004; Moodie et al., 2007; Young et al., 2011; Haneuse & Rotnitzky, 2013; Rytgaard et al., 2022; Ying, 2024b). In contrast, under a static

intervention the subjects are either all treated or all untreated; this is typically the case when considering an average treatment effect (ATE). A second aspect about intervention is 'deterministic' versus 'stochastic'. Deterministic interventions assign a particular value of treatment to each individual, which may depend on the covariates, in the case of a deterministic dynamic intervention. On the other hand, stochastic interventions do not assign a particular value of the treatment, but rather a probability of receiving a particular treatment value (Cain et al., 2010; Díaz & van der Laan, 2012; Díaz & Hejazi, 2020; van der Laan et al., 2018). More specifically, a stochastic intervention assigns treatment randomly based on a probability distribution. Bonvini et al. (2023) describes the example of a Bernoulli distribution for a binary point treatment, where a single parameter $p \in [0, 1]$ indices such an intervention. Finally, an intervention can be 'pre-specified' versus 'otherwise'. For example, Young et al. (2011) described a pre-specified intervention as: "start combined antiretroviral therapy (cART) within 6 months of CD4 cell count first dropping below x or diagnosis of an AIDS-defining illness, whichever happens first". This is a pre-specified dynamic treatment regime as whether an individual starts treatment depends on their own health related history captured by the covariates.

An incremental intervention (Kennedy, 2019) is a special stochastic dynamic intervention. It is not pre-specified, and it intervenes on the observed treatment distribution. Here by observed treatment distribution we mean the default, or baseline, or 'natural', treatment distribution, without the incremental intervention. In addition to the causal interpretation as policy effects above, incremental causal effects have the notable advantage of not requiring the *positivity* assumption typically required of the more traditional causal estimands.

The literature referenced above has studied incremental causal effect in the setting of a binary treatment. For discrete time longitudinal studies, the incremental intervention leads to modifying the odds of receiving the binary treatment at each time point Kennedy (2019); Naimi et al. (2021); Kim et al. (2021); Sarvet et al. (2023). To our best knowledge, however, there is currently no investigation in the literature on incremental causal effects by modifying the distribution of time to treatment. The examples described at the start of this paper fall under time to treatment; that is, given a well defined time zero, how long a subject has to or choose to wait before receiving the treatment.

In this paper, we generalize existing work on incremental causal effects to the time to (initializing) treatment and prove that one can avoid the positivity assumption in this case as well. We first define incremental causal effects by shifting hazard function via hazard ratio. We show that they can be identified by the observed data with only sequential randomization assumption and consistency assumption, without the often needed positivity assumption in causal inference. We construct a generic inverse probability weighted estimator accommodating numerous estimators for the hazard function, coupled with root-$n$ inference results via random weighting bootstrap. Our estimation tools can leverage off-the-shell software from survival analysis. We examine the finite-sample performance of these estimators via extensive Monte-Carlo simulations. We apply our estimator onto a rheumatoid arthritis study to evaluate the effect of methotrexate.

The remainder of the article is organized as follows. In Section 1.1 we review the positivity assumption, and we review related work in Section 2. We introduce notation, the causal estimand, and key assumptions for identification in Section 3.1, where we construct an inverse probability weighting identification approach. In Section 3.2, we propose an inverse probability weighted estimator, together with inferential results yielding confidence intervals. In Section 4.1, we conduct extensive simulations examining finite-performance of our estimator and inference results. In Section 4.2, we apply our framework to answer what happens if we change the probability of receiving methotrexate on rheumatoid arthritis. We end the paper with a discussion in Section 5.

## 1.1 POSITIVITY ASSUMPTION

To identify estimands such as the average treatment effect from the observed data distribution, we typically need at least three common assumptions: consistency, positivity, and no unmeasured confounding or exchangeability (Hernán & Robins, 2020). Consistency is needed in order to bridge the observed data with the potential outcomes, as well as to exclude the possibility of interference where one subject's treatment affects another subject's outcome. No unmeasured confounding has been studied extensively in the literature, including instrumental variables (Angrist et al., 1996; Ying et al., 2019; Ying & Tchetgen Tchetgen, 2023) and proximal approaches (Miao et al., 2018; Tchetgen Tchetgen et al., 2024; Ying et al., 2023), as well as sensitivity analysis towards the violation of

this assumption (Rosenbaum, 2002). Here we focus on the positivity assumption; that is, each subject in the population must have a positive probability of receiving each treatment level. In addition, for estimation approaches such as inverse probability weighting (IPW) or doubly robust approaches to have valid asymptotic inference, the stronger *strict positivity or overlap* assumption is needed; that is, the probability of receiving each treatment level needs to be bounded away from zero. These assumptions, while testable using observed data, can be violated in practice. Bonvini et al. (2023) provided a recividism example where some probationers might be required to obtained the treatment under study by attending behavioral health services, therefore their probability of receiving treatment is one and not receiving treatment is zero. Similar cases can be found for vaccination where some people may not be suitable for receiving the vaccine. D'Amour et al. (2021) showed mathematically that overlap is hard to satisfy when many covariates included in order to meet the unconfoundedness assumption that is also required in many causal inference approaches. For studies involving time-varying treatment, covariates, and outcomes, the positivity assumption is even more of a challenge to meet.

Therefore, many traditional causal quantities of interest might be vetoed by absence of positivity alone after checking the data. On the other hand, incremental causal effects bridge this gap, by not requiring positivity. This is because for subjects with degenerate probabilities (of zero or one) of receiving treatments, perturbing the odds does not change their degenerate probabilities, therefore the positivity can be seen as always satisfied (Kennedy, 2019). More generally speaking, to non-parametrically identify the effect of intervention on the treatment value, such as 0 or 1, requires data on subjects receiving both treatments 0 and 1 over the population of interest. On the other hand, to identify the effect of intervention on the treatment distribution, and not specific values, as will be seen below, does not require data on subjects receiving all treatment values over the population of interest.

## 2 RELATED WORK

As mentioned above Kennedy (2019) proposed for binary treatment incremental intervention which fully resolved the positivity issue mentioned above. A disadvantage of the incremental causal effect, quoting Kennedy (2019), is: "First, we expect incremental causal effects to play a more descriptive than prescriptive role compared to other approaches. Specifically, they give an interpretable picture of what would happen if exposure were increased or decreased in a natural way, but will likely be less useful for informing specific treatment decisions." As will be seen, contrary to what is said here, the incremental causal effect in our setting can yield useful policy-making decisions.

Díaz & van der Laan (2012), Haneuse & Rotnitzky (2013), Díaz & van der Laan (2013), Young et al. (2014), Díaz et al. (2023a), and Díaz et al. (2023b) considered general interventions that depend on the observed treatment process, and are referred to as modified treatment policies or interventions that depend on the natural value of treatment, the latter referring to a treatment that a subject would 'naturally' receive in the absence of it being assigned by an intervention. An example from Haneuse & Rotnitzky (2013) concerns the surgery time for a patient, and asks the question of what if every patient's operating time is reduced by say 5 minutes. Note that each patient's (observed) operating time depends potentially on their individual covariates as well as additional unobserved attributes. In this sense the observed operating time contains more information than what is captured by the measured covariates. In a similar vein, the natural value of treatment also emphasizes the belief that it contains more information than what is captured by the measured covariates (Stensrud et al., 2024). Because these interventions involve more information than the measured covariates, they are different from the pre-specified dynamic intervention introduced in the previous section (Haneuse & Rotnitzky, 2013). Our work here also involves the observed treatment distribution, and considers the continuous time to treatment initialization that has not been previously studied in the literature.

Time to treatment or treatment initialization, or to a large extent similarly, time to treatment termination, has been consider in the dynamic treatment setting (Johnson & Tsiatis, 2005; Yang et al., 2018; Nie et al., 2021). In particular, the first two references (Johnson & Tsiatis, 2005; Yang et al., 2018) considered continuous time to treatment termination, while (Nie et al., 2021) considered discrete time. All of the above are for deterministic interventions.

Continuous time causal inference in longitudinal studies for stochastic interventions, a broader setting than time to treatment initialization studies, is general challenging, and limited work has been

done in the field (Rytgaard et al., 2022; Ying, 2022a; 2024a;b; Sun & Crawford, 2022). Among these, Ying (2022a; 2024a) and Sun & Crawford (2022) only considered static intervention, Ying (2024b) obtained only identification results without any estimation procedure, and none of them (Rytgaard et al., 2022) considered incremental causal effects .

Finally, the literature on causal inference until recently have paid less attention to the positivity assumption than, for example, unmeasured confounding. Petersen et al. (2012), Zhu et al. (2021), and Léger et al. (2022) have recently investigated the importance and implications of this often-overlooked assumption. Hwang et al. (2024) and Matsouaka & Zhou (2024) aimed at relaxing the positivity assumption. Our paper here aims at a new class of estimands which completely avoid positivity, and this in turn opens up more causal questions that one can answer in a study.

## 3 PROPOSED APPROACH

### 3.1 ESTIMAND AND INVERSE PROBABILITY WEIGHTED IDENTIFICATION

Suppose we observe $\{Y, T \wedge \tau, \Delta = \mathbb{1}(T < \tau), L\}$, where $Y$ is an outcome of interest measured at time $\tau$, $T$ is time to (initializing) a certain treatment, and $L$ are baseline covariates. We use $\mathbb{P}$ and $\mathbb{E}$ to represent the distribution of $\{Y, T \wedge \tau, \Delta = \mathbb{1}(T < \tau), L\}$ and expectation with respect to $\mathbb{P}$. We assume $T$ to be absolutely continuous, and write $\lambda(t|l) = \lim_{h \to 0} \mathbb{P}(T < t + h|T \geq t, L)/h$ and $\Lambda(t|l) = \int_0^t \lambda(s|L)ds$ as its hazard function and cumulative hazard function at time $t$ given $L = l$, respectively. As reviewed in the literature above, the hazard function is a natural quantity used to describe the intensity of time to treatment (Yang et al., 2018; 2020). Nonetheless we also write $f(l)$, $f(t|l)$, and $f(y|t,l)$ as the (conditional) densities or probability functions.

We may define the potential outcome $Y_t$ (Neyman, 1923; Rubin, 1974; Holland, 1986) if the treatment were initialized at time $t$. Since $Y$ is measured at time $\tau$, whenever $t \geq \tau$ we denote $Y_t = Y_\tau = Y_\infty$ for someone who is never treated.

Unlike the previous literature, we are interested in answering the question like, "what would be the expected outcome $Y$ if the hazard function $\lambda(t|l)$ were to be doubled?" In general, we may consider shifting the hazard by a positive quantity $\theta(t, l) > 0$. We consider this to be a natural 'hazard preserving' stochastic intervention; that is, to multiply the hazard function for time to treatment by a positive quantity. We note that when $\theta$ is a constant, we have a *proportional hazards* intervention, which has the obvious advantage of simple interpretation. The fact that it coincides with the familiar Cox model is another added advantage. More generally, we might consider recommendation of intensified disease screening for certain high risk groups described via $l$, or after a certain age $t$.

For the rest of the paper we denote $T(\theta)$ as the random draw following the hazard function $\theta(t, l) \cdot \lambda(t|l)$. For example, suppose the time to a certain treatment initialization, $T$, follows the Cox proportional hazards model including a single continuous covariate $L$, with the baseline hazard corresponding to the hazard of a Weibull random variable. Specifically, the hazard function of the time to the treatment initialization at time $t$ given $L = l$ is $\lambda(t|l) = \lambda_0(t) \exp(0.2l)$, where the baseline hazard is $\lambda_0(t) = 0.9t^{0.5}$. Figure 1 shows the effects of shifting the hazard $\lambda(t|l)$ by a constant positive quantity $\theta$, where we plotted the resulting hazard functions and density functions of the time to the treatment initialization $T(\theta)$ under interventions with various constant $\theta$ values.

The corresponding potential outcome under $T(\theta)$ is then by plugging in $T(\theta)$ into $Y_t$ as $Y_{T(\theta)}$. The incremental causal effect is defined as

$$\psi(\theta) = \mathbb{E}(Y_{T(\theta)}). \tag{1}$$

In particular when $\theta \equiv 1$, $\psi(1) = \mathbb{E}(Y_{T(1)}) = \mathbb{E}(Y)$ corresponds to the factual distribution of $T$ that we have observed.

**Assumption 1** (Consistency)**.**
$$Y_{T \wedge \tau} = Y.$$

This assumption links the observed outcome and the potential outcome via the treatment regime. It says that if an individual receives the treatment at time $T$, then his/her observed outcome $Y$ matches $Y_T$. The second assumption is typically referred to as the "sequential randomization" assumption (SRA), "sequential exchangeability", or "no unmeasured confounders" assumption.

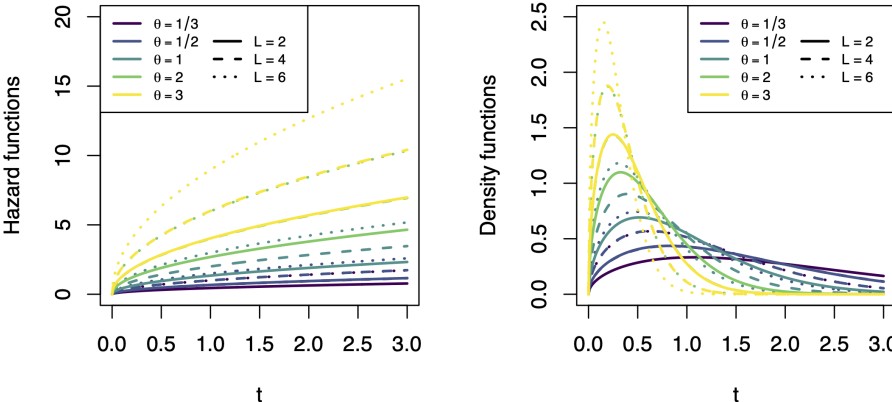

Figure 1: Hazard and density functions of the time to treatment initialization $T(\theta)$ under interventions with various constant $\theta$ values.

**Assumption 2** (No Unmeasured Confounding). *The time to decide treatment and treatment decision is independent of the potential outcomes given the covariate,*

$$T \perp Y_t \mid L, \quad \forall t \in [0, \tau].$$

It requires that time to treatment initialization is determined solely by the collected baseline information $L$.

We prove that Equation (1) can be identified:

**Theorem 1** (IPW Identification). *Under Assumptions 1 and 2, the incremental causal effect $\psi(\theta)$ can be identified by the observed data distribution via*

$$\psi(\theta) = \mathbb{E}(Y_{T(\theta)}) = \mathbb{E}\left\{\theta(T, L)^\Delta e^{-\int_0^{T \wedge \tau}(\theta(t,L)-1)d\Lambda(t|L)}Y\right\}.$$

This identification generalizes the classical IPW identification because the weights $\theta(T, L)^\Delta e^{-\int_0^{T \wedge \tau}(\theta(t,L)-1)d\Lambda(t|L)}$ are the Radon-Nikodym derivatives (or known as likelihood ratio) between the targeted incremental intervention distribution and observed data distribution (both can be found in the appendix), as the classical inverse probability weights.

As noted in the Introduction, while positivity is a fundamental assumption in causal inference for common estimands such as the ATE, here we do not need positivity for identification of the incremental causal effect from the observed data distribution. More specifically, compared with the proof of identification here, traditional pre-specified stochatic dynamic interventions (including deterministic static interventions as special cases) operationalize by replacing $\lambda(T|L)e^{-\int_0^T \lambda(t|L)dt}$ by some pre-specified conditional distribution of $T$. This necessitates to impose an additional positivity assumption. Or in other words, if any person with medical history $L$ can never receive vaccine at time $t$, then we certainly cannot investigate questions like what happens if they do.

### 3.2 INVERSE PROBABILITY WEIGHTED ESTIMATION AND INFERENCE

Suppose we observe a random sample of size $n$ denoted as $\{Y_i, T_i \wedge \tau, \Delta_i = \mathbb{1}(T_i < \tau), L_i\}_{i=1}^n$. Inspired by Theorem 1, if we were to know the truth $\Lambda(t|L)$, then one may plug it in and obtain an estimate of $\psi(\theta)$ by $\frac{1}{n}\sum_{i=1}^n \theta(T_i, L_i)^{\Delta_i} e^{-\int_0^{T_i \wedge \tau}\{\theta(t,L_i)-1\}d\Lambda(t|L_i)}Y_i$. However in practice, $\Lambda(t|L)$ is hardly known to an analyst and must be estimated. Write the estimate as $\hat{\Lambda}(t|L)$, we may define

the plug-in IPW estimator as

$$\hat{\psi}(\theta) = \frac{1}{n} \sum_{i=1}^{n} \theta(T_i, L_i)^{\Delta_i} e^{-\int_0^{T_i \wedge \tau} \{\theta(t, L_i) - 1\} d\hat{\Lambda}(t|L_i)} Y_i.$$

With some regularity conditions on the nuisance estimate $\hat{\Lambda}(t|L)$ given in the appendix, we may derive the following asymptotic results on $\hat{\psi}(\theta)$.

**Theorem 2** (Consistency). *Under Assumptions 1, 2, and the regularity condition that $\hat{\Lambda}(t|L)$ is uniformly consistent for $\Lambda(t|L)$, that is,*

$$\mathbb{P}(\sup_t |\hat{\Lambda}(t|L) - \Lambda(t|L)| > \varepsilon) \to 0$$

*as $n \to \infty$, we have that $\hat{\psi}(\theta)$ converges to $\psi(\theta)$ in probability, that is, for any constant $\varepsilon > 0$,*

$$\mathbb{P}(|\hat{\psi}(\theta) - \psi(\theta)| > \varepsilon) \to 0,$$

*when $n \to \infty$.*

**Theorem 3** (Asymptotic Normality). *Under Assumptions 1, 2, and the regularity conditions that $\hat{\Lambda}(t|L)$ is uniformly consistent and asymptotically linear for $\Lambda(t|L)$, that is,*

$$\mathbb{P}(\sup_t |\hat{\Lambda}(t|L) - \Lambda(t|L)| > \varepsilon) \to 0$$

*and*

$$\hat{\Lambda}(t|L) - \Lambda(t|L) = \frac{1}{n} \sum_{i=1}^{n} \xi_i(t, L) + o_P(1)$$

*for some random process $\xi_i(t, L)$, where $o_P(1)$ converges to zero as $n \to \infty$ uniformly over $t$, then the root-$n$ scaled centered difference $\sqrt{n}\{\hat{\psi}(\theta) - \psi(\theta)\}$ is asymtotically linear and thus converges to a normal variable weakly, that is,*

$$\sqrt{n}\{\hat{\psi}(\theta) - \psi(\theta)\} = \frac{1}{\sqrt{n}} \sum_{i=1}^{n} \Xi_i + o_P(1) \to \mathcal{N}(0, \mathrm{Var}(\Xi_1)),$$

*for some random variable $\Xi$, in distribution.*

We use bootstrap to construct variance estimator of $\hat{\psi}(\theta)$ and build Wald-type confidence intervals.

## 4 EXPERIMENT RESULT

### 4.1 SIMULATION

In this section, we investigate the finite-sample performance of our estimator. We set $\tau = 2$ and generate $n$ i.i.d. copies of $(L_i, T_i, Y_i)$ as follows:

$$L_i \sim \mathrm{Unif}(0, 1),$$
$$\mathbb{P}(T_i > t|L_i) = \exp\{-\exp(0.25L_i)t\},$$
$$Y_i \sim \mathcal{N}(\exp(1 - 1.5L_i - (2 - T_i \wedge 2)), 0.5^2),$$

where we observe $\{L_i, T_i \wedge 2, \Delta_i = \mathbb{1}(T_i < 2), Y_i\}_{1 \le i \le n}$.

We fit a Cox proportional hazard model Cox (1972) to get an estimate $\hat{\Lambda}(t|l)$ because the proportional hazard assumption holds for distribution of $T_i$ given $L_i$. We examine the performance of the IPW estimator $\hat{\psi}(\theta)$ and its variance estimate when $\theta(t, l) \equiv 1/3, 1/2.5, 1/2, 1/1.5, 1.5, 2, 2.5, 3$ by reporting biases, percent biases (%Bias), empirical standard errors (SEE), average estimated standard errors (SD) based on $B = 200$ multiplier bootstrap (van der Vaart & Wellner, 1996; Kosorok, 2008), and coverage probabilities (95% CP) of Wald type 95% confidence intervals using $R = 1000$ simulated data sets of size $n = 200, 1000, 5000$. The results are given in Table 1.

As the simulation results illustrate, $\hat{\psi}(\theta)$ perform well with small biases, across different constant functions $\theta(t, l)$, thus confirming our theoretical results. As expected from theory, variance estimates approach the Monte Carlo variance as sample sizes increase. Similarly, Wald type confidence intervals of $\psi(\theta)$ attain their nominal levels as sample sizes become larger. In the appendix, we consider a more complicated simulation setting.

Table 1: Simulation results of the IPW estimator. We report bias ($\times 10^{-2}$), percent bias (%Bias), empirical standard error (SEE) ($\times 10^{-2}$), average estimated standard erros (SD) ($\times 10^{-2}$) and coverage probability of Wald type 95% confidence intervals (95% CP) of $\hat{\psi}(\theta)$ by $B = 200$ multiplier bootstrap, for $n = 200, 1000, 5000$ sample sizes and $R = 1000$ Monte Carlo samples.

| | $\theta(t\|l) \equiv$ | 1/3 | 1/2.5 | 1/2 | 1/1.5 | 1.5 | 2 | 2.5 | 3 |
|---|---|---|---|---|---|---|---|---|---|
| | $\psi(\theta)$ | 0.957 | 0.893 | 0.808 | 0.694 | 0.404 | 0.336 | 0.297 | 0.273 |
| | Bias | -1.373 | -1.191 | -0.845 | -0.486 | 0.027 | 0.025 | -0.018 | -0.086 |
| | %Bias | -1.435 | -1.333 | -1.046 | -0.7 | 0.067 | 0.074 | -0.061 | -0.315 |
| $n = 200$ | SEE | 7.216 | 6.716 | 6.171 | 5.602 | 4.63 | 4.555 | 4.657 | 4.848 |
| | SD | 6.929 | 6.475 | 5.994 | 5.499 | 4.588 | 4.505 | 4.597 | 4.77 |
| | 95% CP | 93 | 93.1 | 93.4 | 94 | 94.4 | 95.2 | 95 | 94.5 |
| | Bias | -0.396 | -0.388 | -0.253 | -0.139 | 0.031 | 0.043 | 0.039 | 0.013 |
| | %Bias | -0.414 | -0.434 | -0.313 | -0.201 | 0.077 | 0.128 | 0.132 | 0.046 |
| $n = 1000$ | SEE | 3.268 | 3.035 | 2.778 | 2.508 | 2.055 | 2.033 | 2.096 | 2.195 |
| | SD | 3.198 | 2.982 | 2.745 | 2.495 | 2.059 | 2.032 | 2.088 | 2.184 |
| | 95% CP | 94 | 93.9 | 93.7 | 94 | 94.5 | 95 | 95.2 | 94.5 |
| | Bias | -0.069 | -0.109 | -0.035 | 0.001 | 0.022 | 0.021 | 0.021 | 0.001 |
| | %Bias | -0.072 | -0.122 | -0.043 | 0.001 | 0.053 | 0.063 | 0.069 | 0.004 |
| $n = 5000$ | SEE | 1.399 | 1.3 | 1.194 | 1.088 | 0.93 | 0.931 | 0.968 | 1.02 |
| | SD | 1.446 | 1.347 | 1.239 | 1.124 | 0.926 | 0.914 | 0.94 | 0.985 |
| | 95% CP | 95.4 | 95.7 | 95.5 | 95.3 | 94.6 | 94.7 | 94.5 | 94.3 |

## 4.2 EVALUATING INCREMENTAL CAUSAL EFFECT OF METHOTREXATE ON RHEUMATOID ARTHRITIS

We illustrate our framework by estimating incremental causal effects of the anti-rheumatic therapy Methotrexate (MTX) among patients with rheumatoid arthritis, using data from Choi et al. (2002). As a disease-modifying antirheumatic drug, MTX can slow joint damage and the progress of the disease. It is one of the most effective medications for treating inflammatory types of arthritis. This, as well as its long track record and inexpensive price tag, explains why it's usually the first drug prescribed for rheumatoid arthritis. Here we consider the average of reported number of tender joints at one year of follow-up $Y$ as the outcome, an important measure of disease progression.

Methotrexate use was recorded at each monthly clinic visit. At each visit, we classified methotrexate exposure status as ever-treated or never-treated, i.e., once a patient starts methotrexate therapy, he or she was considered on therapy for the rest of the follow-up. Our analysis includes individuals who survived and were followed up more than 12 months, with 1010 patients meeting our inclusion criteria. Figure 2 shows the probability of receiving Methotrexate treatment since patients' first visit to the center, estimated by the Kaplan–Meier estimator $\hat{\mathbb{P}}(T \leq t)$, with its pointwise 95% confidence intervals. As the Figure illustrates, nearly 12% of the patients initiated MTX usage at the first visit and around 25% of the patients were treated at the 12 month visit.

We used a Cox proportional hazards model (Cox, 1972) to estimate the conditional cumulative hazard function of the time to initialize Methotrexate $\Lambda(t|L = l)$, given the following variables $L$: age, sex, past smoking status, education level, rheumatoid arthritis duration, rheumatoid factor positive, health assessment questionnaire, patient's global assessment, erythrocyte sedimentation rate, prednisone use, and number of tender joints (measured at baseline). The proportional hazards assumption was rigorously assessed in the appendix and determined to be valid.

We consider estimands $\psi(\theta)$ by letting $\theta$ being constant functions and varying $\theta \in [0.2, 5]$. These are the incremental causal effects when the hazard rate for initiating MTX at any given time is scaled by $\theta$ across all patients. Such an analysis can illustrate how varying levels of aggressiveness or conservatism in prescribing MTX might influence average disease progression.

Plugging the estimated conditional cumulative hazards $\hat{\Lambda}(t|L)$ into our IPW estimator, we estimated the incremental treatment effect curve with $\hat{\psi}(\theta)$, where we also computed the pointwise 95% confidence intervals (with $B = 200$ multiplier bootstraps). The results are shown in Figure 3. If the hazards function of the time to Methotrexate treatment initialization were increased proportionally for all individuals, the average number of tender joints at one year of follow-up would drop. More

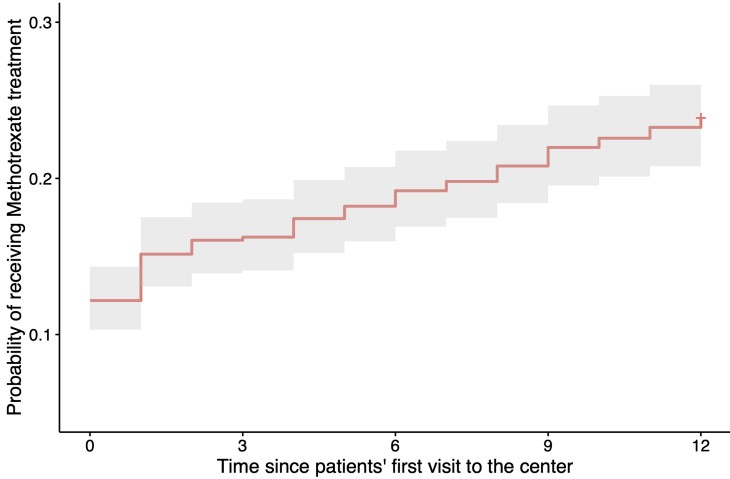

Figure 2: Estimated probability of receiving Methotrexate treatment since patients' first visit to the center $\hat{\mathbb{P}}(T \leq t)$, with pointwise 95% confidence intervals.

specifically, the average number of tender joints at one year of follow-up would decrease by 1.23% from 1.90 if the hazards doubled ($\hat{\psi}(2)$=1.87, 95% CI: 1.81–1.93), and by 5.42% if the hazards were multiplied five-fold ($\hat{\psi}(5)$=1.79, 95% CI: 1.71–1.88). Conversely, the average number of tender joints would increase by 0.83% if the hazards were halved ($\hat{\psi}(0.5)$=1.91, 95% CI: 1.86–1.97), and by 1.51% if the hazards were reduced five-fold ($\hat{\psi}(0.2)$=1.92, 95% CI: 1.86–1.98). The decline in the average number of tender joints as the hazards ratio increases from 0.2 to 5 is consistent with the protective effect of Methotrexate found in Tchetgen Tchetgen et al. (2024), where the joint effects of Methotrexate use at baseline and month six on average of tender joints at month 12 of follow-up were evaluated under a marginal structural linear model and both IPW least squares and proximal recursive least-squares yielded results suggesting a protective effect of Methotrexate on disease progression.

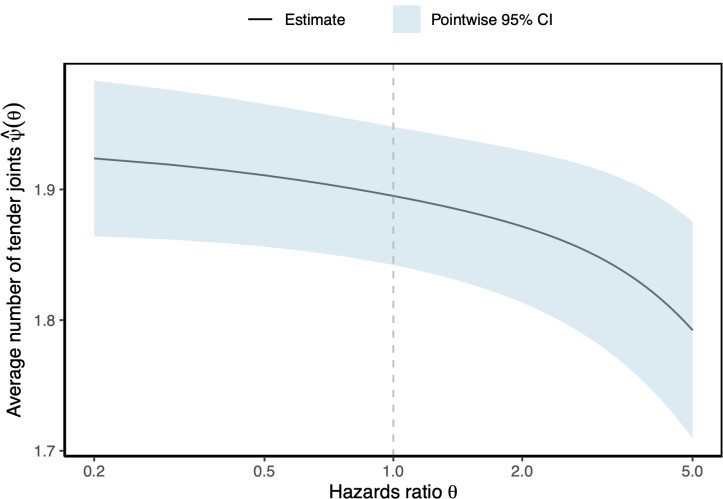

Figure 3: Estimated incremental causal effects $\hat{\psi}(\theta)$, if the hazards were multiplied by factor $\theta$, with pointwise 95% confidence intervals.

## 5 CONCLUSION

In this paper, we extended the concept of incremental causal effects to continuous-time treatment initiation. By defining these effects through shifting the hazard function via the hazard ratio, we demonstrated that they can be identified using only sequential randomization and consistency, without requiring the positivity assumption. We developed a flexible inverse probability weighted estimator, accommodating various hazard function estimators, and provided root-$n$ inference. The finite-sample performance of our framework and its potential real data value were assessed.

There are many open questions. First, one can consider augmenting the inverse probability weighted estimator through semiparametric theory to attain efficiency, using nonparametric estimation for nuisance parameters. This is currently under our investigation. Second, the idea can also be extended to complex longitudinal studies, including time-varying confounders and survival outcomes, or more generally, marked point processes (Rytgaard et al., 2022), càdlàg processes (Ying, 2022a). Third, one might consider when Assumption 2 fails but either instrumental variable assumption (Ying et al., 2019; Ying & Tchetgen Tchetgen, 2023; Ying, 2022b) or proxy variable assumption (Miao et al., 2018; Tchetgen Tchetgen et al., 2024; Ying et al., 2022; 2023) holds. Last, conditional incremental effect (McClean et al., 2024) can also be of interest.

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

## A  PROOFS

### A.1  PROOF OF THEOREM 1

We have

$$
\begin{aligned}
\psi(\theta) &= \mathbb{E}(Y_{T(\theta)}) \\
&= \mathbb{E}[\mathbb{E}\{\mathbb{E}(Y_{T(\theta)}|T(\theta), L)|L\}] \\
&= \int \mathbb{E}(Y_t|T(\theta) = t, L = l)f(l)\{\theta(t|l)\lambda(t|l)e^{-\int_0^t \theta(s|l)\lambda(s|l)ds}\}^\delta \\
&\quad \{\mathbb{E}(Y_\tau|T(\theta) \geq \tau, L = l)e^{-\int_0^\tau \theta(t|l)\lambda(t|l)dt}\}^{1-\delta}dtdld\delta \\
&= \int \mathbb{E}(Y_t|L = l)f(l)\{\theta(t|l)\lambda(t|l)e^{-\int_0^t \theta(s|l)\lambda(s|l)ds}\}^\delta \\
&\quad \{\mathbb{E}(Y_\tau|L = l)e^{-\int_0^\tau \theta(t|l)\lambda(t|l)dt}\}^{1-\delta}dtdld\delta \\
&= \int \mathbb{E}(Y_t|T = t, L = l)f(l)\{\theta(t|l)\lambda(t|l)e^{-\int_0^t \theta(s|l)\lambda(s|l)ds}\}^\delta \\
&\quad \{\mathbb{E}(Y_\tau|T \geq \tau, L = l)e^{-\int_0^\tau \theta(t|l)\lambda(t|l)dt}\}^{1-\delta}dtdld\delta \\
&= \int \mathbb{E}(Y|T = t, L = l)f(l)\{\theta(t|l)\lambda(t|l)e^{-\int_0^t \theta(s|l)\lambda(s|l)ds}\}^\delta \\
&\quad \{\mathbb{E}(Y|T \geq \tau, L = l)e^{-\int_0^\tau \theta(t|l)\lambda(t|l)dt}\}^{1-\delta}dtdld\delta \\
&= \mathbb{E}\left\{\theta(T, L)^\Delta e^{-\int_0^{T \wedge \tau}(\theta(t,L)-1)d\Lambda(t|L)}Y\right\}.
\end{aligned}
$$

where the first equation follows by definition, the second by iteration of expectation, the third by definition of expectation, the fourth by the definition of $T(\theta)$, the fifth by Assumption 2, the sixth by Assumption 1, and the last by the Radon-Nikodym Theorem.

Another intuition of checking this is through densities: The observed data density is

$$
f(Y, T, L) = f(L)\{\lambda(T|L)e^{-\int_0^T \lambda(t|L)dt}f(Y|T, L)\}^\Delta\{e^{-\int_0^\tau \lambda(t|L)dt}f(Y|T > \tau, L)\}^{1-\Delta}.
$$

The target incremental intervention density is

$$
f(Y, T, L) = f(L)\{\theta(T|L)\lambda(T|L)e^{-\int_0^T \theta(t|L)\lambda(t|L)dt}f(Y|T, L)\}^\Delta\{e^{-\int_0^\tau \theta(t|L)\lambda(t|L)dt}f(Y|T \geq \tau, L)\}^{1-\Delta}.
$$

The density for the potential outcomes is

$$
f(Y_{T(\theta)}, T, L) = f(L)\{\theta(T|L)\lambda(T|L)e^{-\int_0^T \theta(t|L)\lambda(t|L)dt}f(Y_T|L)\}^\Delta\{e^{-\int_0^\tau \theta(t|L)\lambda(t|L)dt}f(Y_\tau|L)\}^{1-\Delta}.
$$

Integrating $Y_{T(\theta)}$ yields the parameter of interest (1). By Assumptions 1 and 2, the density for the potential outcomes is equal to the target incremental intervention density. Therefore, to identify (1) through the observed data density, one can use Radon-Nikodym weights to reweight $Y$.

### A.2  PROOF OF THEOREM 2

We have

$$
\begin{aligned}
&\left|\hat\psi(\theta) - \psi(\theta)\right| \\
&= \left|\frac{1}{n}\sum_{i=1}^n \theta(T_i, L_i)^{\Delta_i}e^{-\int_0^{T_i \wedge \tau}(\theta(t,L_i)-1)d\hat\Lambda(t|L_i)}Y_i - \psi(\theta)\right| \\
&\leq \left|\frac{1}{n}\sum_{i=1}^n \theta(T_i, L_i)^{\Delta_i}\left\{e^{-\int_0^{T_i \wedge \tau}(\theta(t,L)-1)d\hat\Lambda(t|L_i)} - e^{-\int_0^{T_i \wedge \tau}(\theta(t,L_i)-1)d\Lambda(t|L_i)}\right\}Y_i\right| \\
&\quad + \left|\frac{1}{n}\sum_{i=1}^n \theta(T_i, L_i)^{\Delta_i}e^{-\int_0^{T_i \wedge \tau}(\theta(t,L_i)-1)d\Lambda(t|L_i)}Y_i - \psi(\theta)\right| \\
&\leq \left|\frac{1}{n}\sum_{i=1}^n \theta(T_i, L_i)^{\Delta_i}\left\{e^{-\int_0^{T_i \wedge \tau}(\theta(t,L_i)-1)d\hat\Lambda(t|L_i)} - e^{-\int_0^{T_i \wedge \tau}(\theta(t,L_i)-1)d\Lambda(t|L_i)}\right\}Y_i\right| + o_P(1).
\end{aligned}
$$

Note that

$$\left| e^{-\int_0^{T_i \wedge \tau}(\theta(t,L_i)-1)d\hat\Lambda(t|L_i)} - e^{-\int_0^{T_i \wedge \tau}(\theta(t,L_i)-1)d\Lambda(t|L_i)} \right|$$

$$= \left| e^{-(\theta(T_i \wedge \tau,L_i)-1)\hat\Lambda(T_i \wedge \tau|L_i)+\int_0^{T_i \wedge \tau}\hat\Lambda(t|L_i)d\theta(t,L_i)} - e^{-(\theta(T_i \wedge \tau,L_i)-1)\Lambda(T_i \wedge \tau|L_i)+\int_0^{T_i \wedge \tau}\Lambda(t|L_i)d\theta(t,L_i)} \right|$$

$$\leq \left| \left\{ e^{-(\theta(T_i \wedge \tau,L_i)-1)\Lambda(T_i \wedge \tau|L_i)+\int_0^{T_i \wedge \tau}\Lambda(t|L_i)d\theta(t,L_i)} + o_P(1) \right\} \right.$$
$$\left. \left\{ |\theta(T_i \wedge \tau, L_i) - 1| \cdot \left| \hat\Lambda(T_i \wedge \tau|L_i) - \Lambda(T_i \wedge \tau|L_i) \right| + \int_0^{T_i \wedge \tau} \left| \hat\Lambda(t|L_i) - \Lambda(t|L_i) \right| d\theta(t,L_i) \right\} \right|$$

$$\leq C \sup_t \sup_{L_i} \left| \hat\Lambda(t|L_i) - \Lambda(t|L_i) \right|.$$

### A.3 PROOF OF THEOREM 3

By a simple Taylor expansion, we have

$$\hat\psi(\theta) - \psi(\theta) = \frac{1}{n}\sum_{i=1}^n \Xi_i(\delta) = \frac{1}{n}\sum_{i=1}^n \theta^{\Delta_i}\hat{\mathbb{P}}(T_i > t|L_i)^{\theta-1}|_{t=X_i}Y_i + \frac{\partial\psi(\theta)}{\partial\Lambda(t)}\xi_i(t,L)|_{t=X_i} - \psi(\theta),$$

and therefore by central limit theorem we have reached the conclusion. $\qquad\square$

## B METHOTREXATE DATA PROCESSING

The raw dataset included the information of 1240 patients. For all the patients, we extracted 11 baseline covariates described in Section 4.2, as well as $T$, the time to receive Methotrexate (MTX) treatment, and the number of tender joints at month $\tau = 12$, which is the outcome $Y$ of interest. Some patients exited the study before reaching month $\tau = 12$, i.e., one year of follow-up, resulting in no recorded $Y$ for these individuals. After excluding them, 1010 patients met our inclusion criteria.

Denote $L_{1,i}$ to $L_{11,i}$ as the baseline covariates, $T_i$ as the time to receive MTX, and $Y_i$ as the number of tender joints at one year of follow-up for each patient ($i = 1, 2, \ldots, 1010$). Define $\Delta_i = \mathbb{1}(T_i < \tau)$. For the $i$th patient, the individual data consists of $\{Y_i, T_i \wedge \tau, \Delta_i, L_{1,i}, \ldots, L_{11,i}\}$.

For constructing the Cox proportional hazards model, we tested the proportional hazards assumption by examining the Schoenfeld residuals and cumulative martingale residuals. For simplicity, we used age_0, sex, smoke_0, edu_0, duration_0, rapos, haqc_0, gsc_0, esrc_0, onprd2_0, and jc_0 as the abbreviations for the covariates age, sex, past smoking status, education level, rheumatoid arthritis duration, rheumatoid factor positive, health assessment questionnaire, patient's global assessment, erythrocyte sedimentation rate, prednisone use, and number of tender joints (measured at baseline), respectively. Figure 4 gives the Schoenfeld residuals plots of time-dependent Cox regression and the results of Schoenfeld individual tests and global Schoenfeld test. Table 2 shows the results of the test of proportionality for fitting an Cox-Aalen model. Figure 5 gives the cumulative martingale residuals plots for each covariate. The results show that the Cox proportional hazards model we constructed is appropriate.

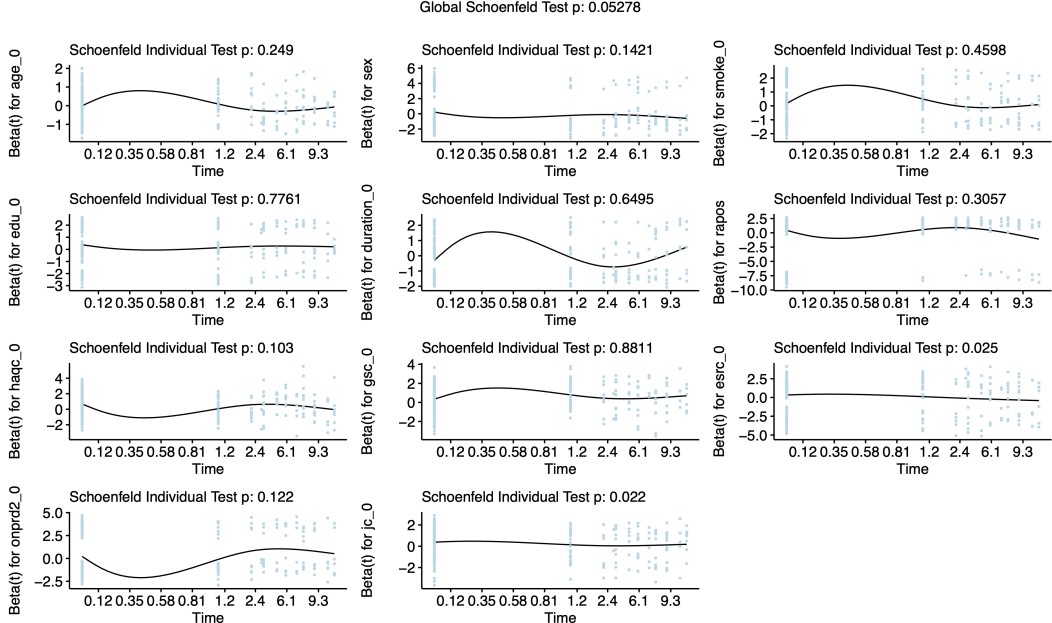

Figure 4: Results of Schoenfeld residuals examination.

Table 2: Results of the test of proportionality for fitting an Cox-Aalen model.

|  | $\sup |\hat{U}(t)|$ | p-value |
|---|---|---|
| age_0 | 16.80 | 0.158 |
| sex | 6.30 | 0.248 |
| smoke_0 | 10.30 | 0.406 |
| edu_0 | 5.52 | 0.932 |
| duration_0 | 10.60 | 0.306 |
| rapos | 4.68 | 0.294 |
| haqc_0 | 9.52 | 0.320 |
| gsc_0 | 6.28 | 0.774 |
| esrc_0 | 8.40 | 0.078 |
| onprd2_0 | 7.98 | 0.140 |
| jc_0 | 14.00 | 0.094 |

## C ADDITIONAL SIMULATION

In this section, we investigate the finite-sample performance of our estimator under a more complicated scenario than that in Section 4.1, including more covariates $L$ and a complex function $\theta(t, l)$. We set $\tau = 2$ and generate $n$ i.i.d. copies of $(L_i = (L_{1i}, L_{2i}, L_{3i}), T_i, Y_i)$ as follows:

$$L_{1i} \sim \text{Unif}(0, 1),$$
$$L_{2i} \sim N(0.5, 0.25^2),$$
$$L_{3i} \sim \text{Bernoulli}(0.5),$$
$$\mathbb{P}(T_i > t | L_i) = \exp\{-\exp(0.1L_{1i} + 0.05L_{2i} + 0.1L_{3i})t\},$$
$$Y_i \sim \mathcal{N}(\exp\{1 - (0.6L_{1i} + 0.3L_{2i} + 0.6L_{3i}) - (2 - T_i \wedge 2)\}, 0.5^2),$$

where we observe $\{L_i, T_i \wedge 2, \Delta_i = \mathbb{1}(T_i < 2), Y_i\}_{1 \leq i \leq n}$. The covariates we consider include a bounded uniform distribution, an unbounded normal distribution and a discrete Bernoulli distribution.

We fit a Cox proportional hazard model Cox (1972) to get an estimate $\hat{\Lambda}(t|l)$ because the proportional hazard assumption holds for distribution of $T_i$ given $L_i$. We examine the performance of the

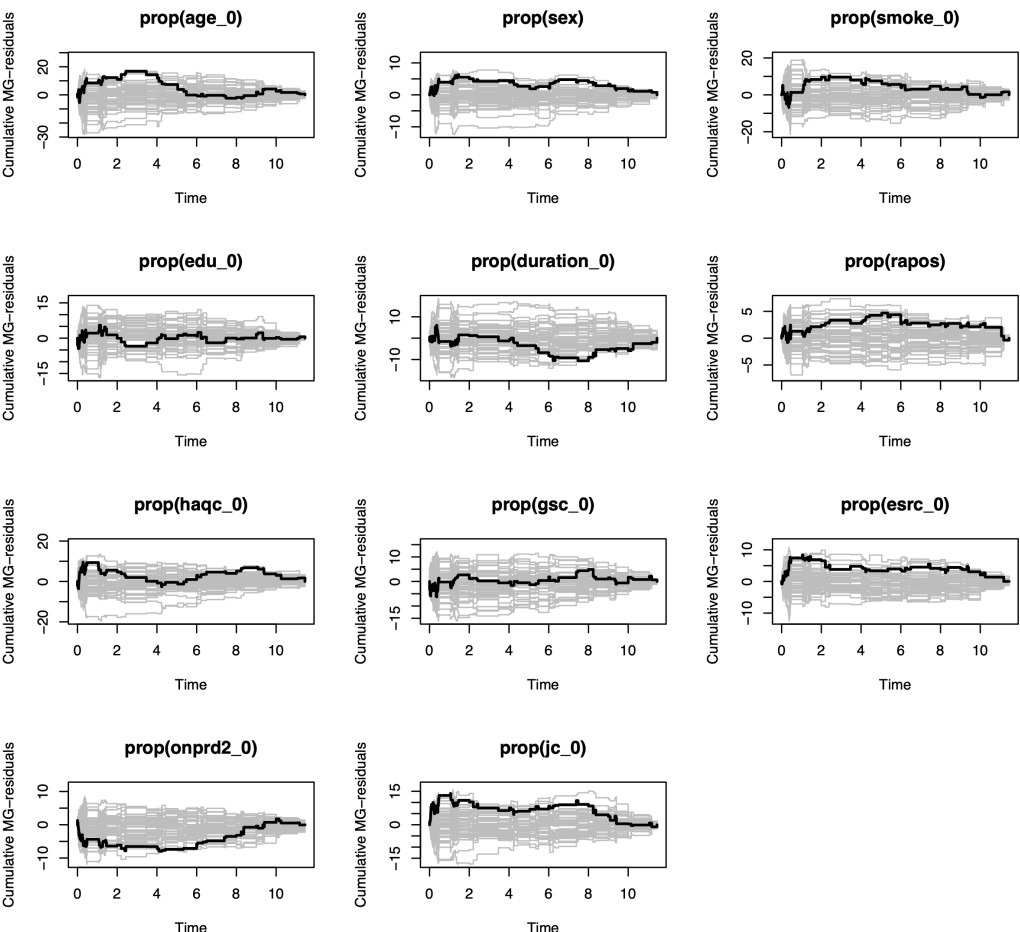

Figure 5: Cumulative martingale residuals plots for each covariate.

IPW estimator $\hat{\psi}(\theta)$ and its variance estimate when $\theta(t,l) = \exp(\beta' l)$ for $\beta \in$

$$\{\beta_1 = (0.1, 0.1, 0.1), \beta_2 = (0.2, 0.2, 0.2), \beta_3 = (0.5, 0.5, 0.5),$$
$$\beta_4 = (0.1, 0.2, 0.5), \beta_5 = (0.1, 0.5, 0.2), \beta_6 = (0.2, 0.1, 0.5),$$
$$\beta_7 = (0.2, 0.5, 0.1), \beta_8 = (0.5, 0.1, 0.2), \beta_9 = (0.5, 0.2, 0.1)\},$$

a much more complicated form. We report biases, percent biases (%Bias), empirical standard errors (SEE), average estimated standard errors (SD) based on $B = 50$ multiplier bootstrap (van der Vaart & Wellner, 1996; Kosorok, 2008), and coverage probabilities (95% CP) of Wald type 95% confidence intervals using $R = 5000$ simulated data sets of size $n = 200, 1000, 5000$. The results are given in Table 3.

As the simulation results illustrate, $\hat{\psi}(\theta)$ perform well with small biases, across different functions $\theta(t,l)$, thus confirming our theoretical results. As expected from theory, variance estimates approach the Monte Carlo variance as sample sizes increase. Similarly, Wald type confidence intervals of $\psi(\theta)$ attain their nominal levels as sample sizes become larger.

Table 3: Simulation results of the IPW estimator. We report bias ($\times 10^{-2}$), percent bias (%Bias), empirical standard error (SEE) ($\times 10^{-2}$), average estimated standard erros (SD) ($\times 10^{-2}$) and coverage probability of Wald type 95% confidence intervals (95% CP) of $\hat{\psi}(\theta)$ by $B = 50$ multiplier bootstrap, for $n = 200, 1000, 5000$ sample sizes and $R = 5000$ Monte Carlo samples.

| | $\theta(t,l) =$ | $\beta_1{}'l$ | $\beta_2{}'l$ | $\beta_3{}'l$ | $\beta_4{}'l$ | $\beta_5{}'l$ | $\beta_6{}'l$ | $\beta_7{}'l$ | $\beta_8{}'l$ | $\beta_9{}'l$ |
|---|---|---|---|---|---|---|---|---|---|---|
| | $\psi(\theta)$ | 0.474 | 0.436 | 0.348 | 0.424 | 0.408 | 0.426 | 0.405 | 0.413 | 0.408 |
| | Bias | 0.172 | 0.196 | 0.287 | 0.216 | 0.236 | 0.201 | 0.233 | 0.225 | 0.219 |
| | %Bias | 0.364 | 0.449 | 0.823 | 0.509 | 0.579 | 0.473 | 0.576 | 0.545 | 0.537 |
| $n = 200$ | SEE | 4.707 | 4.686 | 4.897 | 4.822 | 4.658 | 4.831 | 4.617 | 4.697 | 4.647 |
| | SD | 4.686 | 4.67 | 4.837 | 4.791 | 4.631 | 4.805 | 4.595 | 4.686 | 4.637 |
| | 95% CP | 94.48 | 94.46 | 94.26 | 94.54 | 94.36 | 94.54 | 94.42 | 94.5 | 94.32 |
| | Bias | 0.084 | 0.052 | 0.106 | 0.075 | 0.074 | 0.058 | 0.065 | 0.054 | 0.046 |
| | %Bias | 0.178 | 0.12 | 0.304 | 0.177 | 0.181 | 0.137 | 0.161 | 0.131 | 0.112 |
| $n = 1000$ | SEE | 2.121 | 2.115 | 2.212 | 2.183 | 2.099 | 2.189 | 2.078 | 2.122 | 2.095 |
| | SD | 2.103 | 2.097 | 2.193 | 2.156 | 2.081 | 2.163 | 2.064 | 2.106 | 2.083 |
| | 95% CP | 94.14 | 94.1 | 94.32 | 93.8 | 93.94 | 93.92 | 94.12 | 94.06 | 94.04 |
| | Bias | 0.036 | -0.003 | 0.04 | 0.016 | 0.016 | -0.001 | 0.008 | -0.002 | -0.009 |
| | %Bias | 0.077 | -0.007 | 0.114 | 0.038 | 0.038 | -0.002 | 0.021 | -0.005 | -0.023 |
| $n = 5000$ | SEE | 0.948 | 0.947 | 0.996 | 0.975 | 0.941 | 0.977 | 0.934 | 0.952 | 0.942 |
| | SD | 0.94 | 0.938 | 0.983 | 0.965 | 0.931 | 0.968 | 0.923 | 0.942 | 0.932 |
| | 95% CP | 93.92 | 94 | 94.18 | 94.14 | 94 | 94.1 | 94 | 93.94 | 93.96 |

