# OpenReview forum: "Incremental Causal Effect for Time to Treatment Initialization"
_ICLR.cc/2025/Conference — ICLR 2025 Poster_

### Official Review · Reviewer_roun · 2024-11-01

**Soundness:** 2
**Presentation:** 3
**Contribution:** 2
**Rating:** 6
**Confidence:** 4

**Summary:**

This paper proposes a novel methodology for estimating incremental causal effects in continuous-time settings, with a specific focus on time-to-treatment initiation. This approach intervenes on the intensity (hazard function) of treatment initiation without relying on the positivity assumption. By shifting the hazard function through a multiplicative factor theta, the authors develop an identification strategy using inverse probability weighting. Theoretical justification, along with both synthetic and real-world experiments, is provided.

**Strengths:**

1. The paper addresses a gap in causal inference by extending incremental causal effects to continuous time-to-treatment initiation, an area that has not been extensively studied before.
2. The authors provide consistency and asymptotic normality theorems for their estimand, adding theoretical rigor to their approach.
3. The paper is well-written and easy to follow. I particularly like the related work section which offers a comprehensive background that is beneficial for readers.

**Weaknesses:**

1. A major strength claimed by the authors is that the new estimand avoids the positivity assumption. However, as noted in related work (line 125), "Kennedy (2019) proposed an incremental intervention that fully resolved the positivity issue." Since this paper also focuses on incremental causal effects, this aspect of the contribution may lack novelty.
2. In the synthetic experiment, only a single feature L is used, following a simple uniform distribution. The experiment would be more robust if it included multiple features and more common distributions, such as Gaussian. Additionally, the experiment would benefit from comparisons with other baseline methods, as the paper currently presents only the performance of their model without comparative analysis.
3. In Section 4.2, the authors mention that the decreasing trend in the average number of tenders aligns with findings from a 2002 paper. While it is always impossible to know the true causal evidence in real-world cases, a quantitative comparison of the estimated causal effects with results from other studies would strengthen the paper beyond trend consistency alone.

**Questions:**

See weaknesses.

1. Line 65: The paper states, "In general, the incremental causal effect has the interpretation of a policy effect on the population, instead of the therapeutic effect on an individual." This reminds me the field of reinforcement learning (RL), which is specifically designed for learning policy rewards. Could there be potential benefits in using RL to learn incremental causal effects?
2. Could you provide a concrete example to illustrate the difference between "time to treatment initiation" and "continuous time to initiating treatment"? These terms appear several times in the paper, but their distinction remains unclear to me. Since this paper focuses on the continuous version, clarifying this difference could help motivate the choice.
3. Line 340: typo 25$ should be 25%.

---

> ### Author Response · Authors · 2024-11-18
>
> ## Weaknesses
>
> 1. The novelty claim of avoiding the positivity assumption overlaps with prior work by Kennedy (2019).
>
>    **Response**: Thank you for highlighting this point, which was also noted by other reviewers. To clarify, both the move to continuous time to treatment and avoiding the positivity assumption are novel contributions of this paper. Kennedy (2019) and some other works introduced the concept of "incremental causal effects" in discrete-time settings and demonstrated that it avoids the positivity assumption in that context. Our work extends this concept to continuous time to treatment and proves, in Theorem 1, that the positivity assumption can similarly be avoided in this setting. We will revise the paper to make these contributions more explicit and clearly articulated.
>
> 2. The synthetic experiment lacks feature diversity and baseline comparisons.
>
>    **Response**: We are including additional features following Gaussian and Bernoulli distributions, and also consider different hazard functions and non-constant interventions in the simulation section. This might happen either before discussion period ends or before camera ready if accepted.
>
>    There is no baseline method for estimating the incremental causal effects for comparison, unfortunately. We added percent bias to provide more insight to the performance of our estimator in the simulation section. For the Methotrexate data application, we added a qualitative comparison of the estimated causal effects with those from a 2024 paper.
>
> 3. The real-world comparison relies on trend consistency without quantitative validation.
>
>    **Response**: A qualitative comparison of the estimated causal effects with a 2002 paper focusing on a similar estimand has been added. Since the estimands we used are not exactly the same, a quantitative comparison is not feasible at this time.
>
> ## Questions
>
> 1. Can reinforcement learning (RL) help estimate incremental causal effects?
>
>    **Response**: If the reviewer refers to using RL to learn optimal treatment regimes that maximize a reward function, it is worth noting that learning incremental causal effects through this approach is not straightforward. As discussed in the book *Causal Inference: What If*, when optimizing an outcome-based reward function given \(L\), the optimal treatment choice becomes deterministic. Incremental interventions, being stochastic by nature, would therefore not align with the deterministic optimal regimes learned through RL.
>
> 2. Clarify the distinction between "time to treatment initiation" and "continuous time to initiating treatment."
>
>    **Response**: "Time to treatment initiation" encompasses both "discrete time to initiating treatment" and "continuous time to initiating treatment". We have clarified this by pointing that the 3rd reference Nie et al. (2021) in that paragraph was on discrete time.
>
> 3. Typo: "25$" should be "25%."
>
>    **Response**: Corrected.

---

> ### Author Response · Authors · 2024-11-24
>
> Additional experiments exploring different shift interventions, hazard functions, and more covariates have been added in the appendix.

---

> ### Author Response · Authors · 2024-11-25
>
> I hope you’ve had a chance to review my responses to your comments on our paper. Please let me know if there are any additional concerns.

---

> > ### Comment · Reviewer_roun · 2024-11-26
> >
> > Thank the authors for the detailed response and the additional experiments. I've decided to maintain my current score.

---

### Official Review · Reviewer_sAsU · 2024-11-03

**Soundness:** 3
**Presentation:** 3
**Contribution:** 2
**Rating:** 6
**Confidence:** 3

**Summary:**

This paper studies a novel setting in causal inference: incremental causal effect of intervening the continuous time to treatment initiation by shifting the hazard function. It introduces an IPW estimator with proofs of consistency and asymptotic normality. It is also validated through empirical simulations and a real-world study.

**Strengths:**

1. This paper extends incremental causal effects, which do not rely on the traditional positivity assumption, to a new setting. This advancement allows for new approaches to studying time-to-treatment problems in fields such as public health and policy-making.
2. Theoretical guarantee is provided.
3. The presentation and flow of this paper are clear.

**Weaknesses:**

1. Additional experiments could provide deeper insights into the behavior and robustness of the proposed approach under different scenarios. For example exploring different shift interventions, hazard functions, and comparative analysis against any alternative estimators.

Minor comments:

2. The clarity of Theorems 2 and 3 could be improved by stating all conditions and notations explicitly.
2. In the simulation, it can be made more clear to state the true effect and whether the outcome is censored in the DGP.
3. Typos: in line 59, line 340, and Theorem 3 proof.

**Questions:**

1. If there is unmeasured confounding, is it natural to apply sensitivity analysis or proximal causal inference in this setting? How might these approaches integrate with your proposed IPW estimator?
2. Are the regularity conditions outlined in Theorem 2 and Theorem 3 considered trivial or standard in common survival models?
3. Is there any pattern in the estimator performance (bias, variance, or stability) as $\theta$ changes?

---

> ### Author Response · Authors · 2024-11-18
>
> ## Weaknesses
>
> 1. Additional experiments exploring different interventions, hazard functions, and comparisons could enhance insights.
>
>    **Response**: Thank you for pointing this out, as it was also raised by other reviewers. We will include additional features following Gaussian and Bernoulli distributions, and consider different hazard functions, including Weibull and Rayleigh distributions, as well as non-constant interventions in the simulation section. This might happen either before discussion period ends or before camera ready if accepted, depends on how smooth with the new experiments.
>
>    For comparative analysis, there is no baseline method for estimating the incremental causal effects for comparison, unfortunately. To further evaluate the performance of our estimator in the simulation, we added percent bias to provide more insight. For the Methotrexate data application, we added a qualitative comparison of the estimated causal effects with those from a 2024 paper focusing on a similar estimand.
>
> 2. Theorems 2 and 3 lack clarity due to missing explicit conditions and notations.
>
>    **Response**: Fixed.
>
> 3. The simulation section should explicitly state the true effect and clarify outcome censoring in the DGP.
>
>    **Response**: With interest in the mean outcome $Y_{T(\theta)}$, our estimand is defined as $\psi(\theta) = \mathbb{E}(Y_{T(\theta)})$, which represents the true incremental causal effect when the corresponding hazard is scaled by a constant positive quantity $\theta$. The values of $\psi(\theta)$ have been added to the simulation results in Section 4.1.
>
>    In the data generating process, $(L_i, T_i, Y_i)$ are generated following the distributions listed in Section 4.1. The observations or the data used in our estimator are $(L_i, T_i \wedge 2, \Delta_i = 1(T_i < 2), Y_i)$, where $T_i \wedge 2$ represents the censored time to a certain treatment.
>
> 4. Typos found in line 59, line 340, and Theorem 3 proof.
>
>    **Response**: Fixed.
>
> ## Questions
>
> 1. How can sensitivity analysis or proximal causal inference address unmeasured confounding in this setting?
>
>    **Response**: Sensitivity analysis can indeed be applied in multiple ways. A straightforward approach is to introduce an unmeasured confounder $U$ that affects both treatment $T$ and outcome $Y$ through two separate sensitivity parameters. In such a scenario, the IPW estimator is expected to fail due to the violation of the unconfoundedness assumption. By varying the sensitivity parameters, one can observe how the estimator deviates from the true causal effect. Regarding proximal causal inference, further investigation is required. Specifically, this would involve constructing an IPW bridge function that links the underlying treatment distribution conditioned on $U$ to the target incremental intervention, leveraging appropriate proxies.
>
> 2. Are the regularity conditions in Theorems 2 and 3 standard in survival models?
>
>    **Response**: Yes, they were shown to hold for Cox models (Andersen et al. 2012), and Aalen models (Martinussen et al. 2006). More generally, these conditions were shown to hold for general Givenko-Cantelli and Donsker families (van der Vaart et al. 2013).
>
>    - Andersen, P. K., Borgan, O., Gill, R. D., & Keiding, N. (2012). Statistical models based on counting processes. Springer Science & Business Media.
>    - Martinussen, T., & Scheike, T. H. (2006). Dynamic regression models for survival data. Vol. 1, Springer.
>    - A. W. van der Vaart & Wellner, J. (2013). Weak convergence and empirical processes: with applications to statistics. Springer Science & Business Media.
>
> 3. Is there any observable pattern in estimator performance as \(\theta\) changes?
>
>    **Response**: So far we cannot observe any patterns but thank you for suggesting this. The reason can be that the bias and variance terms across different $\theta$ do not exhibit any clear relations (say monotonicity).

---

> ### Author Response · Authors · 2024-11-24
>
> Additional experiments exploring different shift interventions, hazard functions, and more covariates have been added in the appendix.

---

> ### Author Response · Authors · 2024-11-25
>
> I hope you’ve had a chance to review my responses to your comments on our paper. Please let me know if there are any additional concerns.

---

> > ### Comment · Reviewer_sAsU · 2024-11-25
> >
> > Thank the authors for the detailed responses, which have addressed most of my concerns.

---

### Official Review · Reviewer_cEmo · 2024-11-03

**Soundness:** 3
**Presentation:** 3
**Contribution:** 3
**Rating:** 6
**Confidence:** 3

**Summary:**

This paper extends INCREMENTAL CAUSAL EFFECT to continuous-time treatment. To this end, the author shows that the target quantity is identifiable under certain assumptions, excluding the well-known positivity assumption. An estimator is then proposed that is consistent. The effectiveness of the estimator has been validated through empirical experiments.

**Strengths:**

1- The paper addresses an important problem and proposes an algorithm to solve it.

2- The proposed approach has been analyzed both theoretically and empirically.

**Weaknesses:**

1- Some definitions in Subsection 3.1, such as the hazard function and related concepts, are not clear to the reader. It would be beneficial to provide more detail, as there is still enough space available.

**Questions:**

1- How is it possible to analyze the estimator with finite sample data? For example, is there a high-probability guarantee for it?

I am not completely familiar with the area covered in the paper, and I’m uncertain about its contribution to the field. I may revise my score after considering the feedback from other reviewers.

---

> ### Author Response · Authors · 2024-11-18
>
> ## Weaknesses
>
> 1. Definitions in Subsection 3.1, such as the hazard function, are unclear and lack sufficient detail.
>
>    **Response**: Added.
>
> ## Questions
>
> 1. Can the estimator be analyzed with finite sample data, and are there high-probability guarantees?
>
>    **Response**: High-probability results, as you mentioned, are typically derived in the asymptotic regime where $n \to \infty$. These asymptotic results were investigated in Section 3.2. In statistical practice, it is standard to evaluate estimators' finite-sample performance separately through finite-sample analysis to demonstrate practical performance (e.g., bias, variance) as in Section 4.

---

> ### Author Response · Authors · 2024-11-25
>
> I hope you’ve had a chance to review my responses to your comments on our paper. Please let me know if there are any additional concerns.

---

> > ### Comment · Reviewer_cEmo · 2024-11-25
> >
> > Thank you for your response. After reviewing other feedback, I’ve decided to maintain my current score.

---

### Official Review · Reviewer_rVjv · 2024-11-08

**Soundness:** 3
**Presentation:** 1
**Contribution:** 2
**Rating:** 5
**Confidence:** 3

**Summary:**

In many settings, whether or not a subject receives a treatment at any given time point may be a function of their covariates.  In these settings, we can reason about the time to treatment from when an individual becomes treatment-eligible to when they actually receive treatment.  Such a model can allow us to reason about how changes to covariates can affect time to treatment and, through treatment, some relevant outcome.  This model has the benefit of not requiring the positivity assumption.  The authors define this model in terms of hazard functions and provide an estimator.  They then assess their model on both synthetic and empirical data and demonstrate how it can be used to inform policy/medical practice decisions.

**Strengths:**

While there are some clarity issues with the narrative of the paper, the individual sections and descriptions in the paper are clear and easily readable.  The literature review on incremental causal effects and time to treatment is very thorough, situating the paper nicely in the literature.  The examples chosen, especially the rheumatoid arthritis example, are strong, and the analysis of the rheumatoid arthritis experiment (the reasoning about doubling the hazard decreasing joint pain) is especially compelling and highlights very nicely how this method could be used in practice.

**Weaknesses:**

The biggest weakness of this paper in my eyes is that the problem being solved is not clearly defined.  Some of this seems to be due to language issues (while the occasional grammatical issue or awkward phrase don't generally impede understanding, there are a few parts where the intended meaning isn't clear), and some of this is due to the lack of a clear motivating example".  Specifically:

- In the introduction, the authors define an incremental intervention as an intervention "that is not pre-specified, but rather a function of the observed treatment distribution."  Without defining what is meant by "the observed treatment distribution", I assume that it means which units in the sample population received treatment and which didn't (P(T)), or maybe the conditional probability distribution in the sample population (P(T|L)).  I'm having a hard time understanding what this means, even after having read the whole paper.  The example in this section, as well as the MTX use case in the experiments, seem to deal with an intervention on treatment (such as being assigned to a behavioral health program or being prescribed MTX) that is, presumably, informed based on the individual's covariate values.  So this is an intervention that is a function of the observed covariates, not "of the observed treatment distribution."  Am I misunderstanding something about your approach here?

The organization of the introduction seems a bit backwards to me.  The example in the third paragraph (probationers being assigned to behavioral health services) is a great motivating example for the general "time to treatment initialization" problem setting.  The first paragraph contains two good examples, but the narrative about "time to treatment" is not made clear.  For example, the first 4 sentences of the paper talk about a tech team struggling to keep up with review requests and asks the reader to consider the effect of doubling the number of reviewers on the processing time of requests.  Coming into this paper with causality in mind, this sounds an awful lot like reasoning about intervening on a treatment (the number of reviewers) and measuring the effect on an outcome (the processing time).  However, as becomes clear later in the paper, the "processing time" is not, in fact, the outcome, but the time until treatment.  And if that's time until treatment, then I suppose treatment = somebody reviewing a request.  But then I'm not sure what the outcome is....backlog size??

If you want to open with that example, you should start by clearing explaining how it maps to your problem. An example flow, assuming I'm understanding the problem correctly: "We're interested in understanding how long it takes people on a tech team to respond to review requests.  The time until review is not static, but depends on many features, such as how many reviewers the help desk has at that time.
 After a system outage, the number of requests has increased, creating a large backlog.  The scheduler wants to decrease the size of this backlog, which they plan to do by decreasing the time until review.  The number of reviewers has a large effect on the time until review, so the scheduler decides to double the number of reviewers, which then doubles the likelihood of a request being reviewed at any given time, as requests are often selected for review at random.  This process - reasoning about percentage changes to covariates to determine their effect on time to treat and, thus, the effect of treatment - is called "incremental causal effects"."

The second paragraph of the introduction also oddly placed.  Digging into the different types of interventions is interesting, and the distinctions brought up are quite relevant, but without having a clear problem statement yet, it's unclear how to fit your proposed method into that framework.  I think you're missing a paragraph in the introduction where you define (not technically, but in straightforward language) your problem statement. (i.e., the time until treatment for each subject is based on some measured covariates; the outcome is an effect of that treatment and starts be recorded as soon as treatment is applied to that subject; we want to reason about how changes in the probability of treatment function affect outcome).

Section 1.1 is focused on, and named after the positivity assumption, and highlights bypassing the positivity assumption as a key advantage of incremental causal effects.  However, this section, from what I can tell, never actually explains how it bypasses positivity.  (Also, the phrase "avoids the positivity" is weird - reword that) It's only the introduction, so I don't expect an in-depth explanation yet, but given that it's a whole section in the intro about positivity, at least a sentence giving an intuition about why we can ignore positivity would help.  Following on from that about positivity, it looks like it's addressed in the first paragraph of the Related Work section.  However, the explanation in the related work is not very clear or detailed (and again, especially given how prominently positivity was just highlighted in the introduction, I expected a deeper/clearer explanation).

Some terminology explanation is missing.  Line 171 defines $\lambda(t|l)$ and $\Lambda(t|l)$ as just "its hazard function and cumulative hazard function at time t given L = l, respectively."  I assume "its" here refers to T.  However, from what I can see, you never actually define either hazard function, despite them being fairly core to your method.

I like the setups chosen for both the synthetic and empirical experiments, but the lack of a baseline makes interpreting the results near-impossible.  For example, in the simulation results, you say that your results illustrate that the incremental causal effects "perform well with small biases."  I'm struggling to see how you came to that conclusion from Table 1 alone.  Looking at the numbers in the "Bias" row, they look low, but are they actually low for that problem?  Did you use additional visualizations to come to the conclusion that these numbers represent good performance?

Between the clarity issues throughout and the difficulties in interpreting the experimental results, I don't feel comfortable voting for acceptance.  If these issues are adequately addressed, though, I'm open to increasing my score.

**Questions:**

Can you explain which pieces of this work are novel/provide a contributions list?  From the introduction, it looks like the move to continuous time to treatment is new.  However, the abstract makes it sound like avoiding the positivity assumption is also novel, while the paper makes it seem like that was something that was already shown as a property of incremental causal effects.

I'm not used to seeing L chosen as the variable for covariates/potential confounders. (I've seen W, X, V, C....) It's not a problem, but is the choice of L based on any particular subset of the literature?

As per my confusion in the Weaknesses section, can you clarify what you mean by an incremental intervention being one that is "a function of the observed treatment distribution"?

In the second paragraph of the introduction, you state that, in a static intervention, "the subjects are either all treated or all untreated", in contrast to a dynamic intervention, where treatment could depend on covariates.  This makes it sound like the entire population is either all treated, or all not treated.  This is a valid scenario to consider (e.g., a state government policy that then affects everyone living in that state), but you describe static interventions as being "typically the case when considering an average treatment effect (ATE)", in which case, you typically need examples of both treated and untreated subjects.  I would assume that you meant "each subject is either treated or untreated, assigned independently of their covariates", but that's not particularly close to what you wrote, so I must be misunderstanding something.  Actually, reading the abstract of Bonvini et al (2021), they describe ATE as relating to "the effect of everyone deterministically receiving versus not receiving treatment" - as in, the counterfactual question.  Is that what you're referring to here?

Especially in medical examples (such as the MTX arthritis example), individual covariates can change over time.  Does your model take into account that an individual's covariates L could change over the timesteps before they get treated, which could in turn affect the probability of treatment?

I'm not following the explanation at the beginning of Related Work about how incremental intervention avoids making the positivity assumption.  Summarizing Kennedy (2019), you say that, for subjects with 0 or 1 probability of treatment, we can see positivity as always satisfied "because perturbing the odds does not change their degenerate probabilities".  How does that follow?

In the line before Theorem 1, it says "We prove that ([1]) can be identified".  What is ([1]) referring to here?  Are you referring to Theorem 1?  Assumption 1?  Equation 1?

Are there any baselines you can use for comparison in the experimental results?  Some other effect estimation method, or at the very least some naive baseline that could provide some calibration for the experimental results?

---

> ### Author Response · Authors · 2024-11-18
>
> ## Weaknesses
>
> 1. The problem being solved is unclear, with ambiguous terminology like "observed treatment distribution" and a lack of clear motivating examples.
>
>    **Response**: We have defined the observed treatment distribution and reworded "function of" to "intervenes on."
>
> 2. The introduction's organization is confusing, with examples and problem statements not clearly mapped.
>
>    **Response**: Thank you for highlighting this potential confusion. In fact, to sharp the focus and eliminate confusion, we decided to remove the tech company example as it is not as good as other examples. But still, to clarify, let’s consider the example of social media platforms handling user-reported content, such as comments flagged for "hate language" or "sexually explicit" content. In this case, the outcome could vary depending on the context: a short-term outcome might be the number of other users who see the flagged comment before it is reviewed, while a longer-term outcome could be user satisfaction. Acting quickly and accurately (e.g., removing the flagged content if deemed inappropriate) would reduce the number of users exposed to such comments and likely improve overall user satisfaction.
>
> 3. Opening with examples should better map them to the problem being studied.
>
>    **Response**: We have revised to open with the probationers example from the original third paragraph as suggested.
>
> 4. The second paragraph of the introduction is misplaced and lacks a clear problem statement.
>
>    **Response**: We have rearranged the flow and explicitly stated our problem, emphasizing that the literature has focused on binary treatment, while our contribution is time to treatment in the context of incremental interventions and effects.
>
> 5. Section 1.1 discusses the positivity assumption but does not provide an intuition or explanation for how it is bypassed.
>
>    **Response**: The old Section 1.1 contained multiple topics, including the problem statement and organization of the paper. We have now rearranged it to focus solely on the positivity assumption, added intuition about how incremental effects bypass positivity, and provided a more detailed explanation after Theorem 1.
>
> 6. Core terminology like "hazard function" and "cumulative hazard function" is introduced without proper definitions.
>
>    **Response**: Added.
>
> 7. The lack of a baseline in experiments makes interpreting results difficult, and biases are not clearly contextualized.
>
>    **Response**: We added percent bias to provide more insight into the performance of our estimator in the simulation section.

---

> ### Author Response · Authors · 2024-11-18
>
> ## Questions
>
> 1. Which aspects of this work are novel, and how do they align with existing properties of incremental causal effects?
>
>    **Response**: Thank you for highlighting this point, which was also noted by other reviewers. To clarify, both the move to continuous time to treatment and avoiding the positivity assumption are novel contributions of this paper. Kennedy (2019) and some other works introduced the concept of "incremental causal effects" in discrete-time settings and demonstrated that it avoids the positivity assumption in that context. Our work extends this concept to continuous time to treatment and proves, in Theorem 1, that the positivity assumption can similarly be avoided in this setting. We will revise the paper to make these contributions more explicit and clearly articulated.
>
> 2. Why was $L$ chosen to represent covariates, instead of more typical variables like $W$, $X$, or $C$?
>
>    **Response**: We are following notation from causal inference in medical/health studies where $W$, $X$, $C$ are usually saved for weights, censored event time, and censoring time where time-to-event can also be of interest, one of the future directions of this paper. $V$ can be a valid option but we are using $L$ following the book: Miguel A Hernán and James M Robins. *Causal Inference: What If*. Boca Raton: Chapman & Hall/CRC, 2020.
>
> 3. Can you clarify what it means for an incremental intervention to be "a function of the observed treatment distribution"?
>
>    **Response**: We have clarified and reworded that.
>
> 4. In the second paragraph of the introduction, does "all treated or all untreated" mean counterfactual reasoning where the entire population is treated versus untreated?
>
>    **Response**: Yes, to the very last question. For example, at the conclusion of a randomized clinical trial which estimates the ATE, the clinical guideline going forward would be to treat all patients from that (relevant) population with drug A and not drug B.
>
> 5. Does the model account for time-varying covariates that could influence treatment probabilities?
>
>    **Response**: Thank you for raising this important point. In the current paper, we assume covariates do not change over time, as our focus is on introducing and motivating the concept. However, extending the model to account for time-varying covariates is a natural next step and an avenue for future work. For example, one could consider patient information recorded monthly ($L_k$) and model the hazard function of $T$ between consecutive months. This approach would lead to a new set of inverse probability weights, computed by multiplying the monthly hazard functions up to the time $T$.
>
> 6. Can you clarify how incremental interventions avoid the positivity assumption, especially for degenerate treatment propensities?
>
>    **Response**: Following Kennedy (2019), let us denote $Y$ as the outcome, $A$ as a binary treatment, and $L$ as baseline confounders. Let $P$ and $P^*$ represent the observed and (any) intervened distributions of $(Y, A, L)$, respectively. To infer any functional of $P^*$ (e.g., the ATE), we require that if $P^*(A = a \mid L = l) > 0$, then $P(A = a \mid L = l) > 0$ must also hold. Equivalently, if $P(A = a \mid L = l) = 0$, then $P^*(A = a \mid L = l)$ must also equal 0. The classical positivity assumption ensures this by assuming $P(A = a \mid L = l) > 0$ for all $l$, which is a sufficient condition for this requirement.
>
>    For subjects with degenerate treatment propensity, e.g., $P(A = 1 \mid L) = 0$, where the odds are $P(A = 1 \mid L) / (1 - P(A = 1 \mid L)) = 0$, perturbing the odds as considered in incremental intervention by Kennedy (2019) by multiplying by a factor $\theta$still results in zero odds. This means that the incremental intervention leads to a new distribution $P^*(A = 1 \mid L) = 0$ for these covariates $L$. Hence, the positivity assumption is not needed under incremental interventions.
>
> 7. In the line before Theorem 1, what does "[1]" refer to?
>
>    **Response**: We meant Equation (1) and we have fixed.
>
> 8. Are there any baseline methods for comparison in the experiments?
>
>    **Response**: There is no baseline method for estimating the incremental causal effects for comparison, unfortunately. We added percent bias to provide more insight to the performance of our estimator in the simulation section.

---

> ### Author Response · Authors · 2024-11-25
>
> I hope you’ve had a chance to review my responses to your comments on our paper. Please let me know if there are any additional concerns.

---

### Meta-Review · Area_Chair_baoQ · 2024-12-19

**Metareview:**

The paper introduces a continuous-time version of the "time to treatment" problem, which I found to be particularly relevant in applications such as AI for healthcare and is, to the best of my knowledge, understudied. It can be valuable for the ICLR community to be exposed to this. On the negative side, the original discrete-time formulation covers the bulk of conceptual questions, and it is can be seen from the paper, the technical aspects of the continuous-time version are relatively unsurprising, with the estimation method being a pipeline of standard problems and solutions - usual consistency / convergence in probability / asymptotic normality / bootstrap + Wald CIs. All of this is nice to have, but not the type of result that is the focus of ICLR (but not out of scope either, to be clear). As I've mentioned, the problem itself is still not given as much attention as I think it deserves, and this counts - although one may wonder whether there is some opportunity lost in getting into more subtleties of the problem given that the solution for the given framing is relatively classical - this is also manifested by having quite some space left in the manuscript itself. Reviewers raised all sorts of clarification questions, which I think were valid but also addressable.

Discussion was admittedly underwhelming despite probing, which was disappointing . Reviews were reasonably informative though and, along with the rebuttals, were useful for the assessment. I found the comment about reinforcement learning (reply to reviewer "roun") to be interesting, and I didn't fully understand it: e.g. to which extent the randomness inheriting to this problem clashes with the usual deterministic policies of optimal single-agent problems. Maybe this could be elaborated upon in any particular version of this problem.

**Additional Comments On Reviewer Discussion:**

Discussion was admittedly underwhelming despite probing, which was disappointing . Reviews were reasonably informative though and, along with the rebuttals, were useful for the assessment. I found the comment about reinforcement learning (reply to reviewer "roun") to be interesting, and I didn't fully understand it: e.g. to which extent the randomness inheriting to this problem clashes with the usual deterministic policies of optimal single-agent problems. Maybe this could be elaborated upon in any particular version of this problem.

---

### Decision · Program_Chairs · 2025-01-22

Accept (Poster)